# Patient Safety Culture in a Tertiary Hospital: A Cross-Sectional Study

**DOI:** 10.3390/ijerph20032329

**Published:** 2023-01-28

**Authors:** María Teresa Segura-García, María Ángeles Castro Vida, Manuel García-Martin, Reyes Álvarez-Ossorio-García de Soria, Alda Elena Cortés-Rodríguez, María Mar López-Rodríguez

**Affiliations:** 1Subdirectorate of Nursing, Hospital Universitario Poniente, Servicio Andaluz de Salud, 04700 El Ejido, Almería, Spain; 2Pharmacy Service, Hospital Universitario Poniente, Servicio Andaluz de Salud, 04700 El Ejido, Almería, Spain; 3Intensive Care Unit, Hospital Universitario Poniente, Servicio Andaluz de Salud, 04700 El Ejido, Almería, Spain; 4Public Health and Epidemiological Surveillance Service, Hospital Universitario Poniente, Servicio Andaluz de Salud, 04700 El Ejido, Almería, Spain; 5Department of Nursing, Physiotherapy and Medicine, University of Almería, 04120 La Cañada, Almería, Spain

**Keywords:** adverse events, AHRQ questionnaire, hospital organization, patient safety, safety culture

## Abstract

Patient safety (PS) culture is the set of values and norms common to the individuals of an organization. Assessing the culture is a priority to improve the quality and PS of hospital services. This study was carried out in a tertiary hospital to analyze PS culture among the professionals and to determine the strengths and weaknesses that influence this perception. A cross-sectional descriptive study was carried out. The AHRQ Questionnaire on the Safety of Patients in Hospitals (SOPS) was used. A high perception of PS was found among the participants. In the strengths found, efficient teamwork, mutual help between colleagues and the support of the manager and head of the unit stood out. Among the weaknesses, floating professional templates, a perception of pressure and accelerated pace of work, and loss of relevant information on patient transfer between units and shift changes were observed. Among the areas for improvement detected were favoring feedback to front-line professionals, abandoning punitive measures and developing standardized tools that minimize the loss of information.

## 1. Introduction

According to the World Health Organization (WHO), patient safety (PS) implies cutting down the adverse events related to health care in order to reduce to an acceptable level the risk of patient injury [1]. As a matter of fact, quality of care is based on the basic principle ‘Primun Non Nocere’ which implies that the health care provider is responsible for not causing harm to the patient and for preventing negative outcomes related to the attendance. The occurrence of adverse events due to unsafe care is one of the main causes of death and disability of people in the world [2]. In developed countries, it is estimated that one in 10 patients suffers harm during hospital care [3] caused by adverse events and almost 50% of them can be prevented [4,5]. Efforts to reduce harm during health care to patients can lead to significant financial savings and, most importantly, better patient outcomes [3]. However, health care systems are becoming more complex and several factors such as workspace or electromedical equipment lead to unwanted adverse events [6]. In order to prevent risks and overcome crisis, committed health care providers trained in PS are needed [7]. Thus, professionals capable of generating and solving incidents as well as a culture in PS involving the work environment are crucial points to ensure the success of improvement strategies in health care systems [8].

PS culture is the set of common values and policies shared by all the professionals of an organization. In fact, it is a mental model that understands PS as both an objective and a common commitment to achieve [9]. This culture includes several factors such as health care providers characteristics, institutional features, health care equipment and products, professional protocols, and communication between health care professionals and patients [10]. Some areas of work, such as the nuclear energy industry or aviation, are more developed in this safety culture than the health care field [11]. However, evidence has proved that working on PS leads to better care assistance and patient outcomes [12]. As such, assessing PS culture is a prior action to improve hospital services quality as well as clinical results [13]. In fact, several instruments have been developed to measure PS culture in hospitals, such as the ‘Hospital survey on patient safety culture’ from the Agency for Health care Research and Quality (AHRQ), or the ‘Patient Safety Climate in Health care Organizations’ (PSCHO), developed by the same agency [14,15]. In addition, there are previous studies that have assessed PS culture in Spain, such as the survey applied by the Ministry of Health, Consumption and Social Welfare [16], or the research carried out by the same Ministry and the University of Murcia to measure PS culture in hospitals [17].

Given the importance of PS culture in health care organizations, many authors appeal to understanding and measuring PS culture in order to implement specific improvement measures that are linked to better clinical outcomes [10,11,18]. Thus, following the consensus based on the most current evidence, the measurement of PS culture should be carried out annually [19], since good PS culture in the organization guarantees a decrease in the occurrence of adverse events and its measurement allows determining the circumstances and characteristics of the environment where professionals carry out health care. With the assessment, weaknesses can be found and can be used to implement lines of improvement, in the same way that strengths can help to reinforce and ensure higher-quality care. For all these reasons, this research was carried out in a third-level hospital, belonging to the Andalusian Health System, which cares for a population of 269,864 people [20] and with 291 beds. The aim of this study was to analyze PS culture among the hospital professionals in order to determine the strengths and weaknesses that influence their perception.

## 2. Materials and Methods

### 2.1. Study Design

A cross-sectional and descriptive study was carried out between the months of April and May 2022.

### 2.2. Participants

The sample consisted of health care professionals and non-health care professionals from any unit of the hospital. Permanent and interim professionals, substitutes and personnel in training were included. The exclusion criteria were people involved in the hospital, but who were not in an active situation (temporary disability, paid or unpaid leave, leave of absence or retirement). In order to obtain a representative sample, the GRANMO sample size calculator was used. Thus, with an accuracy of ±5 and a confidence level of 95% in a population of 1763 professionals, it was determined that a minimum of 350 questionnaires were required to detect statistically significant differences.

### 2.3. Instruments

PS culture in the hospital was assessed by the Hospital Patient Safety Questionnaire (SOPS) version 2.0 developed by AHRQ [21], which was translated and validated into Spanish by the Spanish National Health System [22]. This self-administered questionnaire is validated for health care providers and professionals who apply activities to ensure safety in hospitals [23] and it is composed of 12 dimensions and 34 items (Table 1), with positive and negative questions. To obtain the data, the punctuation of the negative questions was inverted [17,18]. Furthermore, although each item was scored on a Likert scale between 1 and 5, responses were recoded into three categories: (1) negative: strongly disagree/never and disagree/rarely; (2) neutral: neither agree nor disagree/sometimes; (3) positive: agree/almost always and strongly agree/always (Table 2).

The punctuation for each dimension of the scale was calculated by applying the following formula:$$\frac{\Sigma Number\;of\;answers\;\left(negative,neutral\;or\;positive\right)to\;items\;in\;a\;dimension}{Number\;of\;total\;answers\;to\;items\;in\;a\;dimension}$$

So, an item or a dimension was considered as strength if: >75% of positive answers (“agree, strongly agree or almost always, always”) to positive questions; >75% negative answers (“disagree, strongly disagree or never, rarely”) to negative questions. On the other hand, an item or a dimension was considered as a weakness if: >50% negative answers (“disagree, strongly disagree, or rarely, never”) to positive questions; >50% positive answers (“agree, strongly agree or almost always, always”) to negative questions [17].

The questionnaire was distributed through the Google Forms platform in order to reach more participants and make it easy to fulfill. The survey included a brief introduction explaining the purpose of this study, the instructions for responding appropriately and an email address for possible queries or doubts. The complete survey was prepared in Spanish language. Likewise, before the questionnaire was available, there were several meetings with members of the hospital’s Patient Safety Committee, the management team, and intermediate positions of the units in order to inform them about the aims of this study. Moreover, the Communication Unit sent mass messages to the workers’ corporate email, and announcements were launched through the hospital’s internal program. The questionnaire was available for one month and reminders were sent every ten days in order to reach the number of responses required.

### 2.4. Data Analysis

Data analysis was performed using IBM SPSS Statistics^®^ for Windows, v.21 and strengths and weaknesses were collected on a report following the criteria scheme proposed by AHRQ. Thus, for the descriptive analysis of sociodemographic characteristics and PS items and dimensions, frequencies and percentages were calculated. Pearson’s chi-square test was used to compare variables between two or more groups. A 95% confidence interval was used. In addition, a reliability test of the dimensions was carried out by calculating Cronbach’s alpha and analyzing the negative responses in a Pareto diagram.

### 2.5. Ethical Considerations

This study was conducted in accordance with the Declaration of Helsinki and the protocol was approved by the Ethics Committee of the Department of Nursing, Physiotherapy and Medicine (EFM 226/2022). In addition, the participants were previously informed of the objective and characteristics of the research, indicating the instructions of the questionnaire, the voluntary nature of their participation, as well as the possibility of withdrawing from this study when they considered it. The anonymous and confidential treatment of the data was guaranteed.

## 3. Results

A total of 350 professionals finally participated in this study. The majority of them were women (73.43%), between 41 and 50 years old (44.84%) and most of them belonged to the hospitalization area (32.57%), followed by external consultations (17.43%), surgical unit and critical care (17.42%), and emergencies (15.42%). Non-care areas accounted for 17.42% of the total number of participants. Considering the professional category, the participation of care technicians stood out (46.29%), followed by nurses (22.28%), specialist doctors (18.57%) and non-assistance professionals (12.85%). 85.70% worked in contact with the patient and 88.85% were professionals without a position of responsibility. 41.14% and 60% indicated that they had been for more than 11 years in the same unit and hospital, respectively. Regarding weekly hours, 69.42% worked 30 to 40 weekly hours (Table 3).

Before analyzing the data from the SOPS questionnaire, and to assess the internal consistency of each dimension, Cronbach’s α (14) was calculated with a result of 0.707 (Table 4).

The analysis of the questionnaire showed that, of the 12 dimensions, ‘Teamwork’ (77.81%), ‘Support given by the manager and the head of unit’ (82.38%) and ‘Communication and responsiveness’ (77.21%) were considered as strengths when obtaining positive scores above 75%. However, no dimension could be considered a weakness since negative scores above 50% were not obtained in any of them. It should be noted that the dimensions ‘Reporting errors’ (71.43%), ‘Information on events related to PS’ (63.50%) and ‘Organizational learning and continuous improvements’ (62.60%) obtained fairly elevated positive scores.

Regarding the items in the questionnaire, it is noteworthy that items A1 (84.29%) and A8 (78.29%) related to teamwork obtained very high positive scores, being considered as strengths. In the same way, items C4 (88.00%) and C6 (78.29%) related to communication and receptivity, as well as item C2 (76.57%) related to communication of errors, obtained indicative scores of strength. Lastly, the three items that make up the dimension ‘Support provided by the manager and the head of the unit’ also obtained positive scores of over 75%. On the other hand, we must highlight item A5 (50.00%), included in the ‘Work pressure and pace’ dimension, as the only one with a negative score above 50% and, therefore, an element to consider as a weakness within the PS.

As for the dimensions, some items obtained quite high scores, although they could not be considered as strengths. Thus, item A9 (70.86%) from one of the dimensions considered as strength (‘Teamwork’) as well as item C7 (73.71%) along with item C5 (68.86%) constituents of another strength dimension (‘Communication and receptivity’), obtained very high positive scores. Likewise, item A4 (68.00%) related to the review of processes, as well as item A14 (64.86%) about the lack of perception of changes that minimize the repetition of errors, yielded quite high positive scores. The same occurred in items C1 (71.43%) and C3 (66.29%) belonging to the ‘Error communication’ dimension. Lastly, it is worth highlighting items D1 (69.43%) and D2 (66.57%) related to the dimension ‘Information on events related to PS’, which also obtained high positive scores (Table 5).

Regarding “Reported incidents”, 48.58% of the participants reported 6 to 10 PS incidents in the last year, and 30.86% reported 1 to 2 incidents. The Mean of reported incidents was 4.63 with a Standard Deviation of 2.67 and a Mode of 7 reported incidents. Finally, the “Qualification of the PS” found positive ratings at 91.71%. This perception of the professionals could be considered a strength (Table 6).

When analyzing the relationship between the dimensions and the sociodemographic variables, we found that the “Teamwork” dimension obtained significant differences according to the care area, with a significantly lower percentage of positive responses in the emergency room (74.07%). In the “Pressure and rhythm” dimension, we found significant differences in relation to the professional category, with the percentage of negative responses being significantly higher in specialist doctors (24.62%). In addition, significantly higher percentages of positive responses were observed in support units (71.67%), in professionals without contact with the patient (68%), and in those with less than one year of work (65.38%). On the contrary, we observed a significantly lower percentage of positive responses in professionals who worked more than 40 h per week (26.58%) (Table 7).

Regarding “Learning and continuous improvement” a significantly higher percentage of positive responses was observed in women (78.21%), in care technicians (80.25%) and in the hospitalization area (89.47%). Additionally, there was a significantly higher percentage of negative responses in professionals who worked more than 40 h (16.46%). In the “Response to errors” dimension, support service areas showed a significantly higher percentage of positive responses (86.67%) and lower in the surgical area and ICU (26.23%) (Table 7).

Likewise, in the dimension “Support from manager and head of unit” a higher percentage of positive responses was observed in the Nursing category and in the hospitalization area. In the “Reporting errors” dimension, significantly higher percentages of positive responses were shown by women (84.05%), care technicians (85.80%), support units (93.33%) and positions of responsibility (94.87%). On the other hand, professionals without contact with the patient reported a significantly lower percentage of negative responses. The “Communication and responsiveness” dimension yielded positive responses in a significantly higher percentage in the hospitalization area (95.61%) (Table 7).

“Reporting events” showed a significantly higher percentage of positive responses among care technicians (81.48%) and a significantly lower percentage of negative responses among professionals without patient contact (2%). In the “Management support” dimension, the non-assistance categories showed a significantly higher percentage of positive responses (68.89%). In addition, higher percentages of positive response were found in support services (85%), professionals without contact with the patient (86%) with a position of responsibility (82.05%) and who had worked in the hospital for 6-10 years (82.50%). The “Transfer and information” dimension had a significantly higher number of positive responses from external consultations (70.49%). Finally, in the PS Qualification, professionals between 1 and 5 years working, showed 100% positive responses (Table 7).

In the Pareto Diagram, the results were that 50% of the negative responses came from three dimensions: pressure and rhythm, response to errors, and information transfer (Figure 1).

## 4. Discussion

This study analyzes PS culture among professionals from a tertiary hospital and provides information on the strengths and weaknesses that influence this perception. Data collection was carried out in all hospital units and by health and non-health professionals, unlike other studies where data are collected only in specific professional categories [24] or health care professionals [25] and in specific units [26]. In this way, the analysis carried out gives an overview that allows us to determine where to invest more effort in PS.

### 4.1. Strengths

Among the best valued strengths was “teamwork”. The consistency of the results with high values in two items, efficient work and help among colleagues, was different between units. These results coincide with studies carried out at the national level [17,25]. In the present analysis, this perception stood out in hospitalization units, which could indicate good coordination between professionals who share tasks throughout the day. Care is planned and carried out by professionals from different categories, producing a feeling of help and efficient work [27]. Likewise, a significant difference was observed with units where the assessment of this dimension is lower. Even so, teamwork is valued positively by professionals, coinciding with a study carried out in emergency departments of Spanish hospitals [28]. A systematic review of the perception of PS culture in hospital settings carried out by Azyabi et al. [19] concluded that teamwork is one of the critical factors that affect PS.

Another strength, according to our results, is the “Support from the manager and head of unit” dimension, highlighting that these professionals take into consideration the staff’s suggestions and that they take measures to solve problems related to PS. The climate of teamwork and the support of management is a parallel circumstance that has a significantly greater relationship in hospitalization units, with the support of managers being mostly manifested in the Nursing category. This coincides with what was described by Kakeman et al. [29], who concluded that nurse leadership and teamwork is associated with the notification of adverse events and, therefore, with a greater PS culture [30].

The dimension related to “Communication and responsiveness” has also been seen as a strength, since the participants affirmed that they could communicate information related to the PS feeling listened to by the people in charge. This dimension also obtained higher scores in hospitalization units. Along these lines, Vicent and Almarberti [10] explain that strategies and interventions must be adapted to different settings within the hospital and the increasing complexity of care, as well as the pressures on hospitals, to provide safe care. Along these lines, some studies consider transparency and active listening essential, focusing on the work of the unit and feedback on their own mistakes [11,27]. Conducting walkthroughs, including formats such as informal hallway conversations, or break room discussions, can increase employee perception that hospital leaders view PS as an important factor, a priority, are committed to safety, and respond to problems identified by those on the front lines [27].

Finally, in the Error communication dimension, there is an item: “When errors are made in this unit, we talk about ways to prevent them from happening again” valued as strength, majority perception expressed by the positions of responsibility and support units. According to these data, attention should be paid so that the incident management report is equitable and with all the base professionals. Communication between health professionals is essential for adequate patient care, leaders should take advantage of the concern and interest expressed by professionals to promote initiatives for the exchange of information and collaboration [31].

All of the above can explain the attitude of learning and continuous improvement, a dimension that, without being considered a strength, showed a high value, especially in technicians. Thus, the participants expressed that there is a review and evaluation of processes. However, they did not appreciate changes in clinical practice that would minimize errors, a circumstance that leads to focusing on the operational framework to make the planned improvements a reality.

The number of incidents reported is among the rates achieved in other studies whose objective is the assessment of PS, finding no differences with those carried out by Abuosi et al. [32] or Wami et al. [33]. One cause of this may be the widespread knowledge that exists in the hospital of the regional incident notification system. Some investigations show that a lack of knowledge of the registration system may be the main cause of underreporting of incidents [34]. The results of this study add to the already existing knowledge that teamwork and the support of those responsible for units found results in an increase in the notification of incidents and improvement of PS [32,35,36,37]. In the same way, the number of incidents reported in this hospital is far from the data of national studies, where the majority of respondents did not report any event related to PS in the last year [17,28].

In general, the PS rating valued by professionals was high. This result is consistent with those carried out in other countries such as Ghana [32], and contrary to other studies where the perception is lower, such as those carried out in Latin American countries [37], southwestern Ethiopia [33], Malaysia [38], Iran [39] and national studies [17,28]. The qualification had a significantly higher percentage of positive responses in professionals with time worked in the hospital between 1 and 5 years. This could be related to greater work experience, greater knowledge about the risks involved in health care and error communication channels [20].

### 4.2. Weaknesses

On the other hand, according to the Pareto diagram, the results of the dimensions ‘Pressure and rhythm’, ‘Response to errors’ and ‘Information transfer’ constituted half of the negative responses of the participants. To this result was added an item belonging to the dimension ‘Pressure and rhythm of work’ that referred to the “Temporary personnel unit” existing in the units. Therefore, it was reported that there are enough staff, but the time and rhythm were not considered sufficient and safe for patient care, highlighting this perception in professionals who work more than 40 h a week. This weakness is significantly greater in doctors, which can be related to the care burden they experience due to the demand for their hiring. Thus, PS culture should focus on the rationality of staffing and is described by several authors as factors to be taken into account since they favor the appearance of errors [17,25,28,34]. This situation invites the need to advise management for the review of staff, or the review of the organization of services in situations of overdemand [27]. Likewise, one must look for causes that condition the pace of work and the perceived stress situation, which can lead to a situation of fatigue and influence patient care [40]. In addition, strategies such as rapid drainage of patients or alternatives to conventional hospitalization should be implemented [36].

Regarding the “Response to errors”, the professionals expressed that when there is an error, they feel judged. However, all errors occur as a consequence of the combination of several factors. To analyze why the error has occurred, the approach must focus on studying the latent conditions and what happened, and not so much on who caused it [30]. Feedback to front-line professionals and dissemination of identified risks should also be encouraged, abandoning punitive measures [18]. Regarding 

Information transfer is another important threat and future line of improvement. The lack of time and loss of information when changing shifts or units, is negatively experienced by professionals. It is thus detected as a weakness that influences the evolution of the patient. This perception has been reflected in other works such as that of Raeissi et al. [39] which suggests that a lack of effective communication between professionals leads to adverse events and negative results in care. This fragmentation in communication must be overcome with coordinated work between units and professionals. According to established recommendations, a line of improvement would be the implementation of a structured communication tool and the training of communication skills in transfers [41].

Finally, in Management support, the responses given by positions of responsibility and professionals between 6 and 10 years old stand out. This could be due to the closer relationship between these groups and management. Thus, we consider it a weakness since a lack of feedback is detected from professionals without responsibility, as well as from professionals with short experience or in specific units [28]. For a closer relationship with front-line professionals, a recommended strategy is the adoption of “walkrounds” systems, since it meets the objectives of proactive and retrospective detection of incidents and opportunities for improvement [42].

## 5. Limitations

Although the present study was carried out using a validated questionnaire and provides novelties in the field of PS, it also has certain limitations that must be mentioned. In this sense, it is considered that the extension of the tool used could have influenced a lower participation. On the other hand, the classification by professionals, with or without responsibility, could have influenced the answers to some questions referring to the support of the heads of unit and direction in the perception of PS. At a global level, a good representation of the different professional categories was achieved, although there were some units with little representation, such as ICU. Because of this significant selection bias, the results must be treated with great caution. It should be taken into account for future research. Finally, limitations of Computer-Assisted Web Interview must be considered.

## 6. Future Research

The results obtained allow us to identify the following strategies to improve the safety climate in the hospital: (1) Promote the strength of teamwork, take advantage of the attitude that exists in hospitalization and extend it to other units. (2) Develop a leadership strategy among medical specialists, extending the feeling that exists in the leadership of Nursing managers. (3) Promote the attitude of active listening and provide support and support for the implementation of lines of improvement as a result of the incident report of PS. (4) Promote good existing communication so that information reaches frontline professionals. (5) Pay attention to the provision and organization of the staff, the rhythms of work and the pressure to which they are subjected. (6) Develop standardized tools that minimize the risks of information loss between units and shift changes. Finally, in a future study, we will consider adding more participants of all services and explore the perceptions of assistance and non-assistance professionals separately.

## 7. Conclusions

This study shows, in general, a high perception of PS among the participants. In the results, there are strengths, but also shortcomings and weaknesses to which attention must be paid.

Among the strengths efficient teamwork, mutual help among colleagues, especially in hospitalization, and the support of the management in the category of Nursing and hospitalization units stand out. The participants acknowledge that the processes are evaluated, and the analysis of adverse events and the search strategies for causes and risks are scored positively. Despite this, they do not perceive that there are objective changes in the processes that minimize the repetition of errors.

Among the weaknesses, the assessment of human resources is compromised by the existence of floating staff and a perception of pressure and accelerated pace of work, especially among doctors and professionals who work more than 40 h a week. Another weakness detected is the loss of relevant information in patient transfer between units and in hospitals shift changes.

The results found invite us to deepen this line of work and develop structured tools that minimize the risks of information loss and the occurrence of PS events. Thus, intervention strategies and efforts should be focused on maintaining the strong points found and prioritizing actions to improve the worst valued ones.

## Figures and Tables

**Figure 1 ijerph-20-02329-f001:**
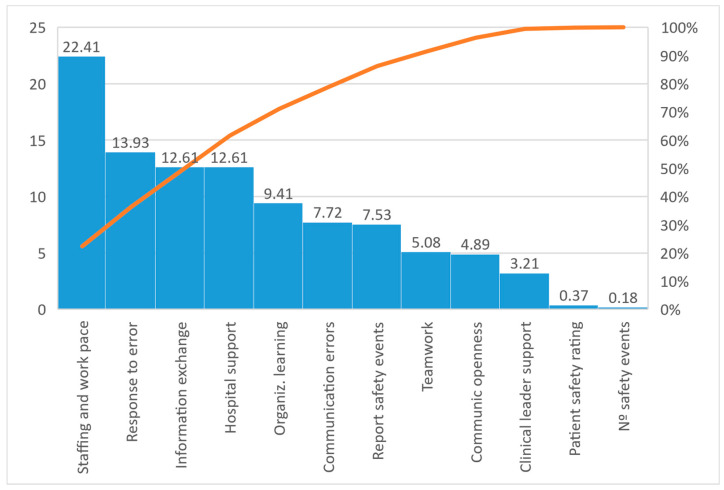
Pareto diagram.

**Table 1 ijerph-20-02329-t001:** SOPS Hospital Survey 2.0 Patient Safety Culture Composite Measures.

Composite Measures	ITEMS
1. Teamwork	A1. In this unit, we work together as an effective team
	A8. During busy times, staff in this unit help each other
	A9. There is an problem with disrespectful behavior by those working in this unit
2. Staffing and work pace	A2. In this unit, we have enough staff to handle the workload
	A3. Staff in this unit work longer hours than is best for patient care
	A5. This unit relies too much on temporary, float, or PRN staff
	A11. The work pace in this unit is so rushed that it negatively affects PS
3. Organizational learning and continuous improvement	A4. This unit regularly reviews work processes to determine if changes are needed to improve PS
	A12. In this unit, changes to improve PS are evaluated to see how well they worked
	A14. This unit lets the same PS problems keep happening
4. Response to error	A6. In this unit, staff feel like their mistakes are held against them
	A7. When an event is reported in this unit, it feels like the person is being written up, not the problem
	A10. When staff make errors, this unit is so rushed that it negatively affects PS
	A13. In this unit, there is a lack of support for staff involved in PS errors
5. Supervisor, manager, or clinical leader support for PS	B1. My supervisor, manager, or clinical leader seriously considers staff suggestions for improving PS
	B2. My supervisor, manager, or clinical leader wants us to work faster during busy times, even if it means taking shortcutsB3. My supervisor, manager, or clinical leader takes action to address PS concerns that are brought to their attention
6. Communication about error	C1. We are informed about errors that happen in this unit
	C2. When errors happen in this unit, we discuss ways to prevent them from happening again
	C3. In this unit, we are informed about changes that are made based on event reports
7. Communication openness	C4. In this unit, staff speak up if they see something that may negatively affect patient care
	C5. When staff in this unit see someone with more authority doing something unsafe for patients, they speak up
	C6. When staff in this unit speak up, those with more authority are open to their PS concerns
	C7. In this unit, staff are afraid to ask questions when something does not seem right
8. Reporting PS Events	D1. When a mistake is caught and corrected before reaching the patient, how often is this reported?
	D2. When a mistake reaches the patient and could have harmed the patient, but did not, how often is this reported?
9. Hospital management support for PS	F1. The actions of hospital management show that PS is a top priority
	F2. Hospital management provides adequate resources to improve PS
	F3. Hospital management seems interested in PS only after an adverse event happens
10. Handoffs and information exchange	F4. When transferring patients from one unit to another, important information is often left out
	F5. During shift changes, important patient care information is often left out
	F6. During shift changes, there is adequate time to Exchange all key patient care information
11. Reporting PS Events	D3. In the past 12 months, how many PS events have you reported?
12. PS Rating	E1. How would you rate your unit/work area on PS

PS: patient safety.

**Table 2 ijerph-20-02329-t002:** Response Options.

Negative		Neutral	Positive	
Strongly disagree	Disagree	Neither agree nor disagree	Agree	Strongly agree
Never	Rarely	Sometimes	Usually	Forever

**Table 3 ijerph-20-02329-t003:** Sociodemographic Data of Participants Characteristics.

		Gender			
		Male		Female	
		Freq	%	Freq	%
Age	20–30 years	3	0.85	19	5.42
	31–40 years	16	4.57	66	18.85
	41–50 years	47	13.42	110	31.42
	>51 years	27	7.71	62	17.71
Assistance area	Surgical area and ICU	20	5.21	41	11.71
	Hospitalization	16	4.57	98	28.00
	External consultations	12	3.42	49	14.00
	Emergencies	23	6.57	31	8.85
	Support services	22	6.28	38	10.85
Professional category	Nursing	22	6.28	56	16.00
	Care technicians	23	6.57	139	39.71
	Specialist doctors	28	8.00	37	10.57
	Non-assistance	20	5.71	25	7.14
Contact with patient	Yes	73	20.85	227	64.85
	No	20	5.71	30	8.57
Responsibility	No, I am a basic professional	77	22.00	234	66.85
	Yes, intermediate charge	16	4.57	23	6.57
Time working in unit	<1 year	7	2.00	45	12.85
	1–5 years	29	8.28	75	21.42
	6–10 years	11	3.14	39	11.14
	>11 years	46	13.14	98	28.00
Time working in hospital	<1 year	3	0.85	7	2.00
	1–5 years	19	5.42	71	20.28
	6–10 years	8	2.28	32	9.14
	>11 years	63	18.00	147	42.00
Work hours per week	<30	4	1.14	24	6.85
	30–40	56	16.00	187	53.42
	>40	33	9.42	46	13.14

Freq.: frequency.

**Table 4 ijerph-20-02329-t004:** Cronbach’s Alpha of Patient Safety Culture Dimensions.

Patient Safety Culture Dimensions	Cronbach’s Alpha If Element Is Deleted *
Teamwork	0.675
Staffing and work pace	0.679
Organizational learning and continuous improvement	0.654
Response to error	0.660
Supervisor, manager, or clinical leader support for patient safety	0.677
Communication about error	0.658
Communication openness	0.673
Reporting patient safety events	0.679
Hospital management support for patient safety	0.671
Handoffs and information exchange	0.681
Number of reporting patient safety events	0.828
Patient safety rating	0.736

(*): >0.6: acceptable; 0.7: good; 0.8: excellent.

**Table 5 ijerph-20-02329-t005:** Dimension and Items Score.

		Negative		Neutral		Positive	
		Freq	%	Freq	%	Freq	%
Teamwork	A1	19	5.43	36	10.29	295	84.29
	A8	25	7.14	51	14.57	264	78.29
	A9	38	10.86	64	18.29	248	70.86
	Total:	7.81%		14.38%		77.81%	
Staffing and work pace	A2	88	25.14	81	23.14	181	51.71
	A3	85	24.24	97	27.71	168	48.00
	A5	175	50.00	103	29.43	72	20.57
	A11	129	36.6	95	27.14	126	36.00
	Total:	34.07%		26.85%		39.07%	
Organizational learning and continuous improvement	A4	53	15.14	59	16.86	238	68.00
	A12	50	14.29	107	30.57	193	55.14
	A14	48	13.71	75	21.43	227	64.86
	Total:	14.38%		22.95%		62.60%	
Response to error	A6	73	20.86	90	25.71	187	53.43
	A7	79	22.57	80	22.86	191	54.57
	A10	54	15.43	68	19.43	228	65.14
	A13	92	25.43	128	36.57	133	38.00
	Total:	21.07%		26.14%		53.26%	
Supervisor, manager or clinical leader support for patient safety	B1	13	3.71	50	14.29	287	82.00
	B2	24	6.86	41	11.71	285	81.43
	B3	14	4.00	43	12.29	293	83.71
	Total:	4.85%		12.76%		82.38%	
Communication about error	C1	38	10.86	62	17.71	250	71.43
	C2	38	10.36	44	12.57	268	76.57
	C3	48	13.71	70	20.00	232	66.29
	Total:	11.64%		14.76%		71.43%	
Communication openness	C4	11	3.14	31	8.86	308	88.00
	C5	36	10.29	73	20.86	241	68.86
	C6	21	6.00	55	15.71	274	78.29
	C7	35	10.00	57	16.29	258	73.71
	Total:	7.35%		15.43%		77.21%	
Reporting patient safety events	D1	41	11.71	66	18.86	243	69.43
	D2	40	11.43	77	22.00	233	66.57
	Total:	11.57%		20.42%		63.50%	
Hospital management support for patient safety	F1	42	12.00	82	23.43	226	64.57
	F2	47	12.43	104	29.71	199	56.86
	F3	112	32.00	98	2800	140	40,00
	Total:	18.81%		27.04%		53.80%	
Handoffs and information exchange	F4	80	22.86	94	26.86	176	50.29
	F5	63	18.00	75	21.43	212	60.57
	F6	58	16.57	94	26.86	198	56.57
	Total:	19.14%		25.05%		55.81%	

Freq.: frequency.

**Table 6 ijerph-20-02329-t006:** Scoring of Patient Safety Rating.

	Negative		Neutral		Positive	
Patient Safety Rating	Freq	%	Freq	%	Freq	%
	2	0.57	27	7.71	321	91.71

Freq.: frequency.

**Table 7 ijerph-20-02329-t007:** Patient Safety Culture Dimension Analysis.

Teamwork			Negative	Neutral	Positive	X²	*p*
Assistance area	Surgical area and ICU	N	1	9	51	20.884	0.007
		%	1.64	14.75	83.61		
	Hospitalization	N	0	5	109		
		%	0.00	4.39	95.61		
	External consultations	N	1	9	51		
		%	1.64	14.75	83.61		
	Emergencies	N	2	12	40		
		%	3.70	22.22	74.07		
	Support services	N	0	4	56		
		%	0.00	6.67	93.33		
**Staffing and work pace**			**Negative**	**Neutral**	**Positive**	**X²**	** *p* **
Professional category	Nursing	N	10	25	43	21.369	0.002
		%	12.82	32.05	55.13		
	Care technicians	N	22	68	72		
		%	13.58	41.98	44.44		
	Specialist doctors	N	16	30	19		
		%	24.62	46.15	29.23		
	Non-assistance	N	4	10	31		
		%	8.89	22.22	68.89		
Assistance area	Surgical area and ICU	N	19	23	19	41.656	0.000
		%	31.15	37.70	31.15		
	Hospitalization	N	10	43	61		
		%	8.77	37.72	53.51		
	External consultations	N	6	30	25		
		%	9.84	49.18	40.98		
	Emergencies	N	11	26	17		
		%	20.37	48.15	31.48		
	Support services	N	6	11	43		
		%	10.00	18.33	71.67		
Contact with patient	Yes	N	48	120	131	10.113	0.006
		%	16.05	40.13	43.81		
	No	N	4	12	34		
		%	8.00	24.00	68.00		
Time working in unit	<1 year	N	2	16	34	13.798	0.032
		%	3.85	30.77	65.38		
	1–5 years	N	20	42	42		
		%	19.23	40.38	40.38		
	6–10 years	N	7	24	19		
		%	14.00	48.00	38.00		
	>11 years	N	23	51	70		
		%	15.97	35.42	48.61		
Work hours per week	<30	N	1	8	19	20.947	0.000
		%	3.57	28.57	67.86		
	30–40	N	35	83	125		
		%	14.00	34.16	51.44		
	>40	N	16	42	21		
		%	20.25	53.16	26.58		
**Organizational learning and continuous improvement**			**Negative**	**Neutral**	**Positive**	**X²**	** *p* **
Gender	Male	N	10	24	59	9.767	0.008
		%	10.75	25.81	63.44		
	Female	N	10	46	201		
		%	3.89	17.90	78.21		
Professional category	Nursing	N	3	19	56	22.122	0.001
		%	3.85	24.36	71.79		
	Care technicians	N	4	28	130		
		%	2.47	17.28	80.25		
	Specialist doctors	N	11	11	43		
		%	16.92	16.92	66.15		
	Non-assistance	N	2	12	31		
		%	4.44	26.67	68.89		
Assistance area	Surgical area and ICU	N	10	18	33	40.638	0.000
		%	16.39	29.51	54.10		
	Hospitalization	N	1	11	102		
		%	0.88	9.65	89.47		
	External consultations	N	4	14	43		
		%	6.56	22.95	70.49		
	Emergencies	N	4	17	33		
		%	7.41	31.48	61.11		
	Support Services	N	1	10	49		
		%	1.67	16.67	81.67		
Work hours per week	<30	N	1	6	21	23.571	0.000
		%	3.57	21.43	75.00		
	30–40	N	6	54	183		
		%	2.47	22.22	75.31		
	>40	N	13	10	56		
		%	16.46	12.66	70.89		
**Response to error**			**Negative**	**Neutral**	**Positive**	**X²**	** *p* **
Assistance area	Surgical area and ICU	N	16	18	27	54.164	0.000
		%	26.23	29.51	44.26		
	Hospitalization	N	3	24	87		
		%	2.63	21.05	76.32		
	External consultations	N	5	14	42		
		%	8.20	22.95	68.85		
	Emergencies	N	5	23	26		
		%	9.26	42.59	48.15		
	Support services	N	2	6	52		
		%	3.33	10.00	86.67		
**Supervisor, manager, or clinical leader support for PS**			**Negative**	**Neutral**	**Positive**	**X²**	** *p* **
Professional category	Nursing	N	0	3	75	12.773	0.047
		%	0.00	3.85	96.15		
	Care technicians	N	2	14	146		
		%	1.23	8.64	90.12		
	Specialist doctors	N	4	4	57		
		%	6.15	6.15	87.69		
	Non-assistance	N	0	5	40		
		%	0.00	11.11	88.89		
Assistance area	Surgical area and ICU	N	4	6	51	21.742	0.005
		%	6.56	9.84	83.61		
	Hospitalization	N	1	5	108		
		%	0.88	4.39	94.74		
	External consultations	N	0	4	57.00		
		%	0.00	6.56	93.44		
	Emergencies	N	1	9	44,00		
		%	1.85	16.67	81.48		
	Support services	N	0	2	58.00		
		%	0.00	3.33	93.67		
**Communication about error**			**Negative**	**Neutral**	**Positive**	**X²**	** *p* **
Gender	Male	N	13	16	64	11.457	0.003
		%	13.98	17.20	68.82		
	Female	N	13	28	216		
		%	5.06	10.89	84.05		
Professional category	Nursing	N	3	14	61	22.016	0.001
		%	3.85	17.95	78.21		
	Care technicians	N	7	16	139		
		%	4.32	9.88	85.80		
	Specialist doctors	N	13	8	44		
		%	20.00	12.31	67.69		
	Non-assistance	N	3	6	36		
		%	6.67	13.33	80.00		
Assistance area	Surgical area and ICU	N	9	14	38	59.922	0.000
		%	14.75	22.95	62.30		
	Hospitalization	N	3	5	106		
		%	2.63	4.39	92.98		
	External consultations	N	2	7	52		
		%	3.28	11.48	85.25		
	Emergencies	N	11	15	28		
		%	20.37	27.78	51.85		
	Support services	N	1	3	56		
		%	1.67	5.00	93.33		
Contact with patient	Yes	N	26	40	233	6.392	0.041
		%	8.70	13.38	77.93		
	No	N	0	4	46		
		%	0.00	8.00	92.00		
Responsibility	No, I am a basic professional	N	25	43	243	6.108	0.047
		%	8.04	13.83	78.14		
	Yes, intermediate charge	N	1	1	37		
		%	2.56	2.56	94.87		
**Communication openness**			**Negative**	**Neutral**	**Positive**	**X²**	** *p* **
Assistance area	Surgical area and ICU	N	0	7	54	23.125	0.003
		%	0.00	11.48	88.52		
	Hospitalization	N	0	5	109		
		%	0.00	4.39	95.61		
	External consultations	N	0	10	51		
		%	0.00	16.39	83.61		
	Emergencies	N	1	13	40		
		%	1.85	24.07	74.07		
	Support services	N	1	3	56		
		%	1.67	5.00	93.33		
**Reporting PS events**			**Negative**	**Neutral**	**Positive**	**X²**	** *p* **
Professional category	Nursing	N	9	18	51	15.128	0.019
		%	11.54	23.08	65.38		
	Care technicians	N	10	20	132		
		%	6.17	12.35	81.48		
	Specialist doctors	N	8	17	40		
		%	12.31	26.15	61.54		
	Non-assistance	N	1	9	35		
		%	2.22	20.00	77.78		
Contact with patient	Yes	N	27	59	213	6.486	0.039
		%	9.03	19.73	71.24		
	No	N	1	5	44		
		%	2.00	10.00	88.00		
**Hospital management support for PS**			**Negative**	**Neutral**	**Positive**	**X²**	** *p* **
Professional category	Nursing	N	7	24	47	19.240	0.004
		%	8.97	30.77	60.26		
	Care technicians	N	8	44	110		
		%	4.94	27.16	67.90		
	Specialist doctors	N	11	25	29		
		%	16.92	38.46	44.62		
	Non-assistance	N	0	14	31		
		%	0.00	31.11	68.89		
Assistance area	Surgical area and ICU	N	12	26	23	49.249	0.000
		%	19.67	42.62	37.70		
	Hospitalization	N	2	32	80		
		%	1.75	28.07	70.18		
	External consultations	N	9	18	34		
		%	14.75	29.51	55.74		
	Emergencies	N	3	22	29		
		%	5.56	40.74	53.70		
	Support services	N	0	9	51		
		%	0.00	15.00	85.00		
Contact with patient	Yes	N	25	101	173	14.435	0.001
		%	8.36	33.78	57.86		
	No	N	1	6	43		
		%	2.00	12.00	86.00		
Responsibility	No, I am a basic professional	N	25	101	185	7.556	0.023
		%	8.04	32.48	59.49		
	Yes, intermediate charge	N	1	6	32		
		%	2.56	15.38	82.05		
Time working in unit	<1 year	N	0	3	7	14.198	0.027
		%	0.00	30.00	70.00		
	1–5 years	N	5	35	50		
		%	5.56	38.89	55.56		
	6–10 years	N	4	3	33		
		%	10.00	7.50	82.50		
	>11 years	N	17	66	127		
		%	8.10	31.43	60.48		
Work hours per week	<30	N	0	11	17	14.286	0.006
		%	0.00	39.29	60.71		
	30–40	N	13	72	158		
		%	5.35	29.63	65.02		
	>40	N	13	24	42		
		%	16.46	30.38	53.16		
**Handoffs and information exchange**			**Negative**	**Neutral**	**Positive**	**X²**	** *p* **
Assistance area	Surgical area and ICU	N	10	20	31	23.272	0.003
		%	16.39	32.79	50.82		
	Hospitalization	N	4	33	77		
		%	3.51	28.95	67.54		
	External consultations	N	3	15	43		
		%	4.92	24.59	70.49		
	Emergencies	N	0	21	33		
		%	0.00	38.89	61.11		
	Support services	N	2	23	35		
		%	3.33	38.33	58.33		
**PS Rating**			**Negative**	**Neutral**	**Positive**	**X²**	** *p* **
Time working in unit	<1 year	N	0	3	7	18.634	0.005
		%	0.00	30.00	70.00		
	1–5 years	N	0	0	90		
		%	0.00	0.00	100.00		
	6–10 years	N	0	2	38		
		%	0.00	5.00	95.00		
	>11 years	N	2	22	186		
		%	0.95	10.48	88.57		

PS: patient safety.

## Data Availability

Not applicable.

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
