# Peer review of "Patient Safety Culture in a Tertiary Hospital: A Cross-Sectional Study"

_ijerph, 2023, doi:10.3390/ijerph20032329_

Round 1

Reviewer 1 Report

Dear Authors.

I think you have written an excellent manuscript. I have some considerations.

- In the introduction, it is interesting to add something about which adverse effects occurs due to not pay attention to patient safety. It highlights the importance of the topic and justify the study.

- After review process you can add the location of the participants.

- In Table 2, you have the negative options in bold, it is better to unify the format. 

- Line 131: "reference" you must change this word into the correct number of references.

- The are categories poorly represented, in a future study you can consider adding more participants of all services. And maybe it is useful to explore the perceptions of assistance professionals and not assistance professionals separate. Have you performed this analysis?

- Line 381: eliminate the point after hospitals.

- Line 385: separate 6. Patents.

- References are not in the journal format.

I hope my comments help you to improve your manuscript. Good job!

Author Response

Dear Authors.

I think you have written an excellent manuscript. I have some considerations.

Firstly, thank you very much for investing your time in reading and reviewing our paper. Below we try to answer to all the considerations you made. We really appreciate every comment and suggestion because they have contributed to improve the quality of the article.

  • In the introduction, it is interesting to add something about which adverse effects occurs due to not pay attention to patient safety. It highlights the importance of the topic and justify the study. >> Thank you for your comment, we have introduced some information about this topic on the introduction section.
  • After review process you can add the location of the participants. >> We will be pleased to add any information in the manuscript. However, we have doubts on this point. Where we must add the location of the participants?
  • In Table 2, you have the negative options in bold, it is better to unify the format. >> Thank you for the comment, we have changed it.
  • Line 131: "reference" you must change this word into the correct number of references. >> Thank you for noticing, it was a mistake.
  • There are categories poorly represented, in a future study you can consider adding more participants of all services. And maybe it is useful to explore the perceptions of assistance professionals and not assistance professionals separate. Have you performed this analysis? >> Indeed, it would be very useful to explore the perceptions of assistance and non-assistance professionals Since this analysis has not been carried out, it has been added as a limitation and future line of research.
  • Line 381: eliminate the point after hospitals.>> Thank you for noticing, we have deleted the point.
  • Line 385: separate 6. Patents. >> Thank you for noticing, we have changed it.
  • References are not in the journal format. >> Thank you for the comment, it has been reviewed.

I hope my comments help you to improve your manuscript. Good job! >> Thank you very much for your comments and your kindness.

Reviewer 2 Report

Instruments:

The Hospital Patient Safety Questionnaire that has been used, in its version 2.0, is developed in the United States. Is it validated for the Spanish population? If so, review who has done it and how they have developed it. If it is not validated for hospitals and the Spanish population, it must be pointed out at work.

Limitations:

The study has a significant selection bias, so it must be highlighted in the limitations, and it must be pointed out that the results must be treated with great caution for this reason.

Author Response

We are very grateful for all the comments as they have helped us to substantially improve our work. Below we try to respond to all your contributions.

Instruments:

  • The Hospital Patient Safety Questionnaire that has been used, in its version 2.0, is developed in the United States. Is it validated for the Spanish population? If so, review who has done it and how they have developed it. If it is not validated for hospitals and the Spanish population, it must be pointed out at work.

The survey is translated and validated into Spanish by the Spanish National Health System (1).

Moreover, in the Agency for Healthcare Research and Quality web you can find the 2.0 spanish version of  Hospital Patient Safety Questionnaire https://search.ahrq.gov/search?q=Hospital+Survey+on+Patient+Safety+spanish. And, it is possible to access the guide to provide a basic understanding of how to administer the SOPS Hospital Survey 2.0, analyze the data and the results. https://www.ahrq.gov/sites/default/files/wysiwyg/sops/surveys/hospital/AHRQ-Hospital-Survey-2.0-Users-Guide-5.26.2021.pdf.

Likewise, the European Union Network for Patient Safety (2), a cooperative body that aims to promote patient safety in the European Union, recommends the use of the Hospital Survey on Patient Safety Culture (HSOPS) questionnaire to measure safety culture.

Several validation studies of the survey in Spanish have been found in different settings:

A study carried out in the surgical environment evaluating the psychometric properties of the Latin American Spanish version of the HSOPS questionnaire to propose a validated tool for use in perioperative settings concludes that the psychometric analyzes provide general support for 9 of the 12 dimensions analyzed in the questionnaire. The findings suggested further validation studies before applying the results only to perioperative settings (3).

Another validation of the questionnaire was carried out specifically with Nursing students. The conclusion was that it is a useful and versatile tool to measure the level and strength of the patient safety culture (4).

Finally, a systematic review (5) identified studies where the SOPS questionnaire was used to see the characteristics of the dimensions and collect data on the safety culture in hospitals. The studies revealed a predominance of underdeveloped or weak hospital organizational cultures in terms of patient safety.

  1. Traducción y validación de la encuesta de la AHRQ para medir la cultura de la seguridad del paciente en atención primaria. Available online: https://seguridaddelpaciente.es/es/proyectos/financiacion-estudios/proyectos-sscc/semfyc/2010/ (accessed on 15 January 2023)
  2. Available online: https://www.eu-patient.eu/projects/completed-projects/eunetpas/ (accessed on 15 january 2023)
  3. Calvache, J.A.; Benavides, E.; Echeverry, S.; Agredo, F.; Stolker, R.J.; Klimek, M. Psychometric Properties of the Latin American Spanish Version of the Hospital Survey on Patient Safety Culture Questionnaire in the Surgical Setting. J Patient Saf. 2021,17(8): e1806-13, doi: 10.1097/PTS.0000000000000644
  4. Ortiz, J.; Orkaizagirre-Gómara, A.; Sánchez, M.; Urcola-Pardo, F.; Germán-Bes, C.; Lizaso-Elgarresta, I. Adapting and validating the Hospital Survey on Patient Safety Culture (HSOPS) for nursing students (HSOPS-NS): A new measure of Patient Safety Climate. Nurse Educ Today. 2019, 75:95-103, doi: 10.1016/j.nedt.2019.01.008
  5. Reis, C.T.; Paiva, S. G.; Sousa, P. The patient safety culture: a systematic review by characteristics of Hospital Survey on Patient Safety Culture dimensions. Int J Qual Health Care. 2018,30(9):660-77, doi: 10.1093/intqhc/mzy080

Limitations:

  • The study has a significant selection bias, so it must be highlighted in the limitations, and it must be pointed out that the results must be treated with great caution for this reason. >> Thanks for pointing this out. This point has been highlighted in the limitations section.

Reviewer 3 Report

The research makes an contribution to the literature. After taking into account a few corrections, the work does not raise any objections.

Materials and Methods

2.1. Study Design

Was this a Computer Assisted Web Interview (CAWI) study?

If so, have the limitations of this testing technique been considered?

2.3. Instruments

In what language was the survey prepared?

2.5. Ethical Considerations

Bioethics committee number - add.

3. Results

The results presented in the table require correction 0,85 correct 0.85

Due to the large number of results, some of the results should be included in the annex.

4. Discussion

Divide the discussion into sections on strengths and weaknesses to keep the results transparent.

References

The list of bibliographic references is no work reference.

35. García-Moran, M.C.; Gil-Lacruz, M. El estrés en el ámbito de los profesionales de la salud. Persona. 2016, 19, 11, doi: 480 10.26439/persona2016.n019.968

Author Response

The research makes a contribution to the literature. After taking into account a few corrections, the work does not raise any objections.

Thank you very much for spending your time reviewing our article and for the recognition to our work. Below, we try to respond to the different comments, and we thank you for your contribution as it has improved our paper.

Materials and Methods

2.1. Study Design

  • Was this a Computer Assisted Web Interview (CAWI) study? Exactly, it was a CAWI study.
  • If so, have the limitations of this testing technique been considered? Thank you for your comment, this limitation has been mentioned in the “limitations” section of the manuscript.

2.3. Instruments

  • In what language was the survey prepared? The survey was prepared in Spanish, and we have clarified this point on the manuscript

2.5. Ethical Considerations

  • Bioethics committee number - add. >> Thank you for noticing, the code has been added on the manuscript.
  1. Results
  • The results presented in the table require correction 0,85 correct 0.85 >> Thank you for the comment, it has been reviewed.
  • Due to the large number of results, some of the results should be included in the annex. >> We will be pleased to include in the annex those results you find necessary.
  1. Discussion
  • Divide the discussion into sections on strengths and weaknesses to keep the results transparent. >> Thank you for the suggestion, the discussion has been divided into sections "strengths" and "weaknesses".

References

  • The list of bibliographic references is no work reference. García-Moran, M.C.; Gil-Lacruz, M. El estrés en el ámbito de los profesionales de la salud. Persona. 2016, 19, 11, doi: 480 10.26439/persona2016.n019.968 >> Thank you for your comment, but if you visit the DOI we present on the reference you could check that it is a journal paper.

Round 2

Reviewer 2 Report

"The survey is translated and validated into Spanish by the Spanish National Health System" it is recommended that it appear in the instruments section; to confirm that it is not only translated into Spanish but also validated.

Congratulations for your work.

Author Response

Thank you very much for spending your time reviewing our article and for the recognition to our work.

In reference to your comment, we have added a sentence to indicate that the scale was translated and validated into Spanish, as you suggested (lines 97-98). In addition, we have added the reference corresponding to this validation (22. Traducción y validación de la encuesta de la AHRQ para medir la cultura de la seguridad del paciente en atención primaria. Available online: https://seguridaddelpaciente.es/es/proyectos/financiacion-estudios/proyectos-sscc/semfyc/2010/ (accessed on 15 January 2023)

Again, thank you very much for your interest and for helping us to improve the work.